# Depression Disorders in Mexican Adolescents: A Predictive Model

**DOI:** 10.3390/children10071264

**Published:** 2023-07-22

**Authors:** Gilda Gómez-Peresmitré, Romana Silvia Platas-Acevedo

**Affiliations:** Faculty of Psychology, The National Autonomous University of Mexico, Av. Universidad 3004 Col Copilco-Universidad, Alcaldía, Coyoacán, México City C.P. 04510, Mexico; romsip@unam.mx

**Keywords:** depression, mood disorder, depressive disorder, depression in high school students, depressive symptomatology

## Abstract

Depression is a type of mood disorder that can impact individuals of any age. A variety of factors, including biological, psychological, and environmental factors, can contribute to the likelihood of developing depression. If the environment in which a person exists does not support its occurrence, the disorder may not manifest. The current research follows a retrospective, correlational approach, utilizing a non-probability sample of 557 high school students from public schools in Mexico City. This sample includes 181 males and 376 females, aged between 15 and 18 years, with an average age of 15.66 and a standard deviation of 0.68. The main objective of this research is to identify the variables that serve as risk factors for the development of depressive disorders in Mexican adolescents in high school. The data show that 78% of the adolescents in the total sample were at risk of depression, which is consistent with what has been reported by other researchers. The regression model shows that alcohol and drug consumption is associated with and influences the emergence and presence of depressive symptomatology and major depressive disorder. Adolescents with different sexual orientations than heterosexuals are twice as likely to suffer depression and emotional dysregulation. It was confirmed that the developmental stage and adolescence contributes as a context that favors the evolution of such a symptomatology.

## 1. Introduction

Depression, a mood disorder, has the potential to impact individuals of all ages. According to the World Health Organization [1], in 2020, symptoms of depression include persistent sadness, irritability, a sense of emptiness, and a loss of interest or pleasure in activities, experienced throughout most of the day, almost every day, for at least two weeks. Additional indications include difficulty concentrating, overwhelming feelings of guilt or diminished self-worth, thoughts of death or suicide, changes in appetite or weight, and feelings of fatigue or reduced energy levels.

The emergence of a depressive disorder can be influenced by various factors, including biological, psychological, and environmental elements that contribute to its development; although it is true that there is a genetic and hereditary part, this disorder may not develop if the environment in which the person is immersed does not favor it. Research conducted on twins indicates that approximately 40% to 50% of the risk for developing depression can be attributed to genetic factors. Similarly, studies involving families demonstrate that individuals with first-degree relatives who have experienced depression are two to three times more likely to develop depression themselves at some stage in their lives, compared to individuals without a family history of the disorder [2].

Historically, it has been seen that women tend to present depression more often than men, with a 2:1 ratio [3]; in addition, gender and life cycle issues that are present in women may increase the probability of developing depression at some point in life. Biological factors, such as the fluctuation of hormones in deciding cycles, including menstruation and menopause, have weight in the appearance of depressive symptoms or emotional reactions to stressful situations in life [4]. The different socio-contextual situations to which women are exposed compared to men should also be considered. Gender roles, physical and sexual abuse, and the entry into the labor field, formerly monopolized by men, which favors unequal power relations [5]. These are some of the daily situations that women face, and from which depressive symptoms can be triggered.

Violence is a perfect example to understand the impact of these social conditions in which some women live and how they affect mental health. Vázquez Machado [6] conducted a study with 64 depressed women and 32 women without any disorder. The results showed that 67.2% of the depressed women had previously (12 months maximum) suffered some episode of violence (physical or psychological), and when comparing the two groups with an odds ratio, researchers discovered that individuals who had experienced recent abuse were 6.17 times more probably to exhibit clinical depression compared to those without such a history.

The cultural structures and lives in which people are immersed from birth have consequences for the way they cope with life. In a study conducted by Rodríguez [7], it was seen that the perception of social support received can contribute considerably to depressive symptomatology. Children who received low social support had much lower positive self-esteem than children who received high social support; the latter not only had higher positive self-esteem but also showed a more positive mood [8].

Socioeconomic status has also been associated with depressive symptomatology. Ferrel et al. [9] studied the relationship between the socioeconomic status of a sample of college students and the level of depression. They concluded that socioeconomic status appears as a fundamental triggering factor of this disease. Rodríguez [7] saw that the low structural support that can be received by lower socioeconomic classes can also generate lower positive self-esteem, positive mood, energy–interest and self-esteem compared to the middle-class population that was studied.

According to another study carried out by Álvarez et al. [10], higher percentages of depression—minimal, moderate, and severe—were seen in children from rural areas rather than in urban children. González et al. [11], in turn, showed that negative characteristics observed at school, with friends, and in the neighborhood function as risk factors for depressive symptomatology, as well, possibly, for poor emotional self-regulation.

Family structure can condition health aspects in such a subtle way that it is impossible to perceive it without a critical eye. Large sectors of the population find themselves in living conditions that lead to stressful situations (economic, family problems, etc.), while they do not realize that they receive any external support to cope with these situations.

Adolescence is presented as a stage where multiple changes occur (physical, psychological, cognitive, identity, etc.). During this transitional phase of adolescence, where individuals undergo numerous personal transformations and encounter external pressures associated with their environment, it is not surprising to observe the emergence of emotional issues. Álvarez et al. [10] mention that the family climate and available resources that support young people are related to depressive symptomatology; thus, within a cohesive, expressive, and organized family, and where all members are encouraged to be independent and close to others, it is exceedingly difficult for emotional maladjustment to occur. This same study showed that there is an inversely proportional correlation among family conflicts, which lead to worse family cohesion and depressive symptoms in adolescents; therefore, the family support provided to adolescents can work in two ways: either as a protective factor that helps to prevent a depressive disorder or as a risk factor by provoking these symptoms.

Eating disorders can be associated with challenges in emotional regulation among adults [12] and adolescents of all genders [13]. Both boys and girls exhibit diverse ways of expressing their emotions, with girls more inclined to internalize positive emotions, whereas boys often externalize their emotions [14]. However, like eating disorders, most studies on emotional dysregulation have primarily focused on mothers and daughters. These studies have found that children of depressed mothers are more prone to using unhealthy strategies to regulate their emotions compared to children of nondepressed mothers [14,15]. Nevertheless, similar to eating disorders, the majority of research investigating emotional dysregulation has predominantly centered on mothers and daughters. These studies have revealed that the offspring of mothers experiencing depression are more susceptible to adopting unhealthy mechanisms for regulating their emotions compared to children of nondepressed mothers [14,15]. Furthermore, it has been observed that in middle childhood and adolescence, the capacity of a child to regulate their emotions seems to be more impacted by their mother’s aptitude for emotional regulation rather than their father’s [16]. Additionally, research indicates that mothers and fathers play distinct roles in shaping their children’s ability to regulate their emotions. They respond in distinct ways when faced with their children’s challenging behaviors, thereby making unique contributions to the development of their children’s emotional regulation skills [17,18].

Another cause of depression is the abuse of addictive substances; between 30% and 50% of psychiatric patients are drug-dependent, and among adolescents with substance abuse, 70% of those surveyed suffer from anxiety and/or depression. The use of alcohol, a substance known as a depressant drug, is used by approximately 40% of young people, mostly males. The relationship between alcoholism and depression does not have a clear cause; it seems that each alcohol could produce certain depressive symptoms and depression can lead to abuse and dependence on these substances [19]. Depression and alcohol consumption in adolescents are intricately connected issues with significant implications for their well-being. There are various underlying factors that contribute to the co-occurrence of depression and alcohol consumption in this population, including genetic predispositions, shared environmental influences (such as family or peer factors), neurobiological factors, and psychological vulnerabilities [20,21].

It is widely seen that depression and alcohol consumption often coexist in adolescents, and research supports a bidirectional relationship between the two. Adolescents experiencing depression may turn to alcohol as a means of coping with their emotional distress. Conversely, excessive alcohol consumption can contribute to the onset or worsening of depressive symptoms in susceptible individuals [22].

Studies have consistently proved that adolescents with depression are at a heightened risk of engaging in problematic alcohol consumption compared to their nondepressed peers. Furthermore, the presence of both depression and alcohol consumption in adolescents is associated with more severe symptoms of both conditions, increased functional impairment, and an elevated risk of developing other mental health disorders [23].

A study carried out by Córdova et al. [24], conducted on 323 participants between 12 and 18 years of age, who were drug users, concluded that depression is a strong predictor of substance abuse in both men and women. The greater the number of depressive symptoms, the greater the severity and intensity of substance use. Risk eating behavior (REB) is a group of abnormal behaviors related to food intake and is associated with social isolation, depressive symptoms, anxiety, substance problems, identity problems, and high impulsivity, among others [19]. 

In a study conducted with 892 students in Mexico, it was shown that depression increases 3.4 times the risk of having moderate REB and 8.1 of suffering from high REB, while depression and figure preoccupation increased the risk of REB by 3.14 for moderate REB and 2.95 times of suffering from high REB [25]. According to the American Psychological Association (APA), sexual orientation refers to a lasting emotional, romantic, or sexual attraction an individual experiences towards others. Sexual orientation exists along a spectrum, ranging from exclusive heterosexuality to various forms of bisexuality [26]. Studies have shown that women and individuals belonging to sexual minorities, which encompass all sexual orientations apart from heterosexuality, are twice as prone to experiencing depression compared to heterosexual men [27]. Some studies [28] refer that these disparities in depression appear to be more pronounced among heterosexual men. Other studies involving LGBT youth show elevated scores of emotional distresses, emotional dysregulation, self-injurious, and suicidal ideation in contrast to heterosexual youth [29].

Other authors [30] suggest that young people resort to self-injury as an emotional response. This is related to a deficiency in the management of proper emotional regulation strategies. According to Klonsky [31], self-injury is identified as a coping mechanism that assists in reducing the intensity of overwhelming negative emotions, such as feelings of sadness and loneliness. Some worldwide surveys report high comorbidity between anxiety and depression disorders [32]. This comorbidity is detected more among adolescents, where between 25% and 50% of young people with depression present, in turn, and symptoms of anxiety [33]. In Mexico, there have been few studies on the evaluation of the variables that are involved in the development and maintenance of depression. The aim of the present study was the identification of risk factors for the development of depression in Mexican adolescents in high school.

## 2. Materials and Methods

### 2.1. Sample

The design of the current study involved a cross-sectional, correlational approach utilizing a non-probability sample of n = 557 high school students (181 males and 376 females) from public schools in Mexico. The participants’ age ranged from 15 to 18 years, with a mean age of 15.66 and a standard deviation of 0.68.

### 2.2. Instrument

In the study, researchers used scales from the Online Test for Self-Screening: Risk Factors of Eating Disorders, Depression, Social Anxiety, and Self-Injury (OTESSED) [34]. The test comprised several sections, starting with a sociodemographic section that gathered information on factors such as age, employment, parents’ educational level, sexual orientation, alcohol consumption at home, and family communication. The next section was the Depression and Suicidal Ideation Scale, which consisted of 11 multiple-choice questions ranging from “never” to “always”. The scale’s reliability coefficient was ϖ = 0.86 (α = 0.91) for men and ϖ = 0.89 (α = 0.92) for women, indicating good internal consistency. The Risk Factors Associated with Eating Disorders Scale (EFRATA) section included 11 items with response options ranging from “never” (1) to “always” (5). A higher score on this scale indicated a greater problem. The reliability coefficient for this section was ϖ = 0.88 (α = 0.71) for males and ϖ = 0.84 (α = 0.82) for females, demonstrating good reliability. The Social Anxiety Scale section consisted of 12 items with multiple-choice responses ranging from “never” to “always” This scale aimed to measure levels of social anxiety, and its reliability coefficients were ϖ = 0.88 (α = 0.90) for men and ϖ = 0.87 (α = 0.87) for women, indicating good internal consistency. The Emotional Dysregulation section assessed the regulation of positive and negative emotions using 14 items. The reliability coefficients for this section were ϖ = 0.75 (α = 0.82) for both men and women ϖ = 0.75 (α = 0.84), indicating acceptable internal consistency. Overall, these scales from the online test demonstrated good to acceptable reliability coefficients across the various sections, allowing for the assessment of self-reported risk factors for eating disorders, depression, and social anxiety.

### 2.3. Procedure

The administrators of multiple high school campuses were approached to request the participation of their students in the study. Students who agreed to participate received a link to the research instrument through email. In order to ensure ethical standards, informed consent was obtained from both the school authorities acting as legal guardians and the participants themselves. The study followed the guidelines outlined in the Helsinki Declaration and adhered to the General Health Law on Research that governs the country where the study took place. As the research was considered to be low risk, it received approval from the ethics committee of the Faculty of Psychology (Psychology (FPCE_08032021_H_AC 27 April 2021).

### 2.4. Statistical Analysis 

The study utilized the Statistical Package for the Social Sciences [SPSS], V. 22 (IBM, MEXICO) for the data analysis. To figure out the variability of the responses obtained, the percentage distribution was analyzed using frequencies. Likewise, the criteria for the normality skewness (symmetry +/−1.5) and kurtosis (+/−2) were obtained for each item considering the criteria suggested by Potthat [35]. The results confirmed that all of the items had a good index of variability and normality in the responses. Pearson’s correlation analysis was conducted to establish the relationships among the variables and to further conduct a linear regression analysis with depression as the dependent variable. Additionally, a binary logistic regression analysis was carried out to evaluate the predictor variables. The AMOS statistical package was used to estimate the effect and relationships among the multiple variables through structural modeling.

## 3. Results

The questionnaire was used to collect sociodemographic information from the participants, supplying information on the population studied. The responses to the data obtained are presented below (See Table 1).

### 3.1. Pearson’s Correlation

To assess the relationships among the study variables, including depression, risky eating behavior, anxiety, emotional dysregulation, sexual orientation, frequency of alcohol and drug consumption at home, and frequency of drug and alcohol use at school, a Pearson’s correlation analysis was conducted.

The results showed moderate correlation coefficients for the variables depression, REB, dysregulation, and anxiety, while for the variable anxiety the correlation was high (0.60) (See Table 2) [36].

Before constructing the structural model, a multiple linear regression model was employed to determine the weight and direction of the independent variables and their impact on predicting the dependent variable. The enter method was utilized, with depression as the dependent variable, and the independent or predictor variables included risky eating behavior, anxiety, dysregulation, sexual orientation, violence, and family factors, as well as drug and alcohol consumption both at home and at school.

The Table 3 below displays the coefficients of determination and multiple correlation coefficients between depression (i.e., dependent variable) and the other independent variables (IVs).

According to the data from the regression model (Enter method), the behaviors of depression, anxiety, dysregulation, family violence, sexual orientation, and alcohol and drug use at home and at school had a coefficient of determination (R^2^ = 0.28).

### 3.2. Logistic Regression

To identify the risk predictors, a binary logistic regression analysis was conducted to assess the impact or influence of the studied variables as predictors of depression. The dependent variable in this analysis was the presence or absence of the risk of depression, while the independent or predictor variables included risky eating behavior, anxiety, family violence, dysregulation, sexual orientation, and alcohol and drug consumption.

As can be seen in Table 4, the variable depression presented a medium effect in its relationship with REB, presenting two times (OR (95% CI) = 2.03 (1.2–3.28)) more probability of this behavior occurring. The OR value of the rest of the variables shows, in each case, a medium effect. For the variable anxiety, it is 3.2 times more likely to occur when there is depression (OR (95% CI) = 3.24 (1.9–5.4)). For the variable emotional dysregulation, there is a 2.9 times greater probability of occurrence (OR (95% CI) = 2.90 (1.7–4.8)) when there is depression. In relation to sexual orientation, there is a 3.6 times greater probability of depression when it is different from a heterosexual orientation (OR (95% CI) = 3.6 (1.8–7.3)). For the variable violence in the family, there is a 0.44 (OR (95% CI) = 0.44 (0.22–0.86)) probability of depression. For the variable alcohol and drugs at school, there is a 0.48 times greater probability that this behavior is related to depression. With regard to alcohol and drugs, there is 0.30 greater probability (OR (95% CI) = 0.30 (0.15–0.61)). The logistic regression model explained 40% of the variance (R^2^ Nagelkerke = 0.40), and it correctly classified 84% of the adolescents with depression, with a higher probability of correct classification in adolescents classified as at risk of depression (95%) than in adolescents without depression (40%).

### 3.3. Structural Model

The structural model predictive for depression showed adequate levels for the goodness of fit (Figure 1), with acceptable *p*-values > 0.95 (CFI, GFI, TLI, and IFI), as well as RMSEA and SMRR < 0.05 [37,38]. The anxiety variable presented a β = 0.60, the emotional dysregulation variable a β = 0.47, REB β = 0.35, and sexual orientation β = 0.25. With respect to the values for the explained variance, the squared correlations reported for the variable anxiety were an R^2^ = 0.36, for the variable depression an R^2^ = 0.27, dysregulation an R^2^ = 0.22, and REB an R^2^ = 0.15.

## 4. Discussion

The main objective of this research was the identification of variables that are risk factors for the development of depressive disorders in Mexican adolescents in high school. After analyzing the data, it was found that 78% of the adolescents in the entire sample were identified as being at risk of experiencing depression. This result aligns with previous studies conducted by Ibrahim et al. [39,40] and other researchers.

Several predictors (anxiety, REB, sexual orientation, and drug use among others) related to the presence of depression in adolescents were considered. The results of the regression model show how drug and alcohol use influence the presence of depression, which has been highlighted in several studies, e.g., Contreras-Olive [41] and Restrepo et al. [42]. Alcohol is the most consumed legal drug in Mexico, and its negative effects have been repeatedly associated with depression, a disease that affects 300 million people worldwide [43]. Pedrelli [23] points out that adolescents who drink alcoholic beverages to excess show a strong association with the presence of depressive symptoms and major depressive disorder. In relation to drug use, Sanchez-Hervas [44] reports a study in which addicted adolescents showed major depression and, in some of them, depressive symptoms prior to drug use.

Adolescence is a critical period in the developmental history of individuals in which they experience biological, behavioral, and social changes that will allow them to reaffirm their personality, self-esteem, and self-awareness and achieve their identity [45]. Authors such as Genise [27], Hatzenbuehler et al. [28], and Marshal et al. [29] point out that adolescents with different sexual orientations than heterosexual are twice as likely to suffer depression, emotional anxiety, and dysregulation. The family bond during this stage is a determinant for healthy growth, and poor family communication can negatively influence the inadequate venting of negative emotions and, therefore, this generates depression [46,47].

According to Caldera-Zamora [48], women demonstrate a higher inclination towards experiencing mental health issues, specifically an increased vulnerability to mood disorders like depression and anxiety [40].

Anxiety is a common trait in patients with eating disorders [49], as noted by Toro and Vilardel [50], who identified anxiety as a product of excessive preoccupation with the body or physical figure. The results of the present study report that anxiety obtained a *B* = 0.60, proving its relationship with depression. Antecedent studies have found anxiety symptomatology as a risk predictor for the development of eating disorders and depression, likewise, in clinical populations, comorbidity with generalized anxiety disorder is one of the most common [51,52,53,54,55].

As could be observed, the presence of REB was statistically significant in the logistic regression analysis, demonstrating that for every depressed adolescent there is a probability of presenting two times more REB, as reported by Nuño et al. [56], Unikel et al. [25], and De la Vega et al. [57], who point out that adolescents, regardless of their chronological age but who do present depression, have a tendency towards REB. Unikel et al. [25] found a strong association between the variable’s depression, REB, and anxiety.

Emotional dysregulation showed in the logistic regression analysis that people with depression are three times more likely to have poor emotional regulation. Gratz and Roemer [58] point out that an individual’s awareness of his or her emotional state is key to their regulation. During adolescence, it is difficult to find and label emotions, particularly when their valence is negative. Poor emotional regulation can lead to various consequences, including poor impulse control, generating higher levels of anxiety and depression, as well as substance use and abuse. These behaviors are strongly associated with eating disorders [59,60,61].

Preventing depression in teenagers is crucial because of its significant effects on their emotional well-being, academic performance, and interpersonal relationships. Various strategies to help prevent depression in adolescents should be considered: encourage open communication through the provision of a safe and nonjudgmental space where teens can freely express their feelings and concerns. Similarly, coping skills should be taught to help teens develop problem-solving skills and positive thinking and to effectively manage stress and life challenges. Lifestyle changes involving healthy eating, physical exercise, as well as quality sleep, will allow adolescents to develop strategies that allow them to deal with adversity in a more functional/adaptive way [62,63]. 

## 5. Conclusions

The research findings revealed that almost 80% of adolescents were at significant risk of experiencing depression. The analyses of regression highlighted an association between alcohol and drug consumption, indicating a predictive factor for depressive symptoms and major depressive disorder. Furthermore, the study concluded that adolescence, being a developmental stage, plays a crucial role in providing a conducive environment for the manifestation of these symptoms. The study also emphasized that the changes experienced during this phase can result in poor emotional management, leading to crises and severe depression.

Additionally, it is crucial to acknowledge that the lack of empathy and support for diverse sexual orientations can elevate the prevalence of depression, particularly when accompanied by familial violence. Hence, it is imperative to consider these factors when evaluating and addressing the mental health of adolescents.

## 6. Strengths and Limitations of the Study

Supplying information that writes down the existence of a high percentage at risk of depression in the adolescent population has a valuable preventive value. However, inclusive management of the environment that favors the origin and maintenance of depression is not foreseen. It is of foremost importance not to forget the great challenge that the stage they are going through imposes on them.

## Figures and Tables

**Figure 1 children-10-01264-f001:**
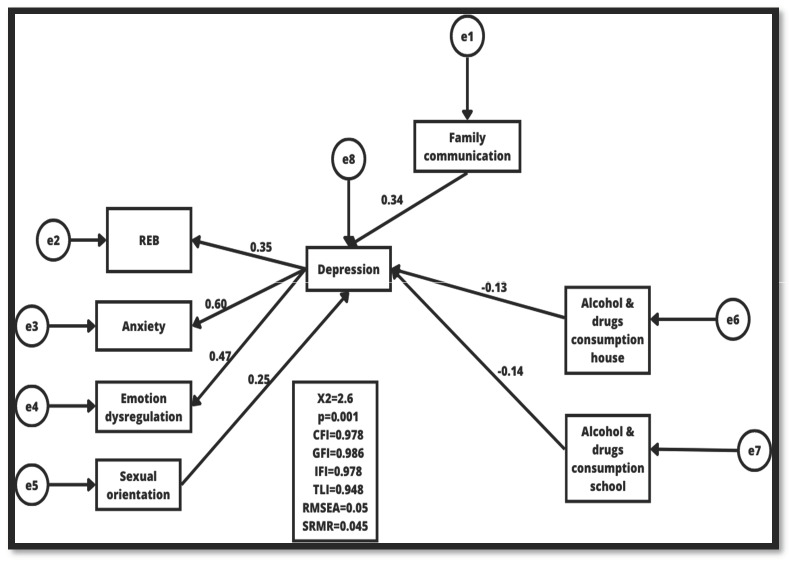
Depression structural model.

**Table 1 children-10-01264-t001:** Sociodemographic data.

	Male (n = 181)	Female (n = 376)
**Living with**		
Nuclear family	135 (74.6%)	294 (79.1%)
Single parent (Mother or father)	45 (24.9%)	80 (21.3%)
Siblings	1 (0.6%)	1 (0.3%)
Partner	-	1 (0.3%)
**Father’s occupation**		
No parent	36 (19.9%)	41 (10.9%)
Informal economic activity	23 (12.7%)	62 (16.5%)
Formal economic activity	121 (66.9%)	253 (67.3%)
Unknown	1 (0.6%)	8 (2.1%)
Unemployed	-	12 (3.2%)
**Father’s education**		
No parent	36 (19.9%)	41 (10.9%)
Basic education	38 (21%)	108 (28.7%)
Middle education	59 (32.6%)	119 (31.6%)
Higher education	48 (26.5%)	106 (28.2%)
No education	-	2 (0.5%)
**Mother’s occupation**		
No parent	8 (4.4%)	5 (1.3%)
Informal economic activity	24 (13.3%)	47 (12.5%)
Formal economic activity	83 (45.9%)	182 (48.4%)
Unpaid job	66 (36.5%)	142 (37.8%)
**Sexual orientation**		
Heterosexual	136 (75.1%)	248 (66%)
Bisexual	17 (9.4%)	99 (26.3%)
Homosexual	16 (8.8%)	7 (1.9%)
Other	12 (6.6%)	22 (5.9%)
**Domestic violence**		
Yes	28 (15.5%)	123 (32.7%)
No	153 (84.5%)	253 (67.3%)
**Use of alcohol and drugs school**		
Consumption	67 (37%)	177 (47.1%)
Nonconsumption	114 (63%)	199 (52.9%)
**Use of alcohol and drugs house**		
Consumption	44 (24.3%)	143 (38%)
Nonconsumption	137 (75.7%)	233 (62%)

**Table 2 children-10-01264-t002:** Pearson’s correlation for the variables REB, anxiety, dysregulation, sexual orientation, family violence, and alcohol and drug use with depression.

Variables	Pearson Correlation
REB	0.40 **
Anxiety	0.60 **
Emotional dysregulation	0.47 **
Sexual orientation	0.33 **
Violence in the family	−0.26 **
Drugs and alcohol at school	−0.20 **
Drugs and alcohol at home	−0.28 **

** Correlation is significant at the 0.01 level (two-tailed).

**Table 3 children-10-01264-t003:** Lineal regression.

Model	R	R Square	R Square Adjusted	Standard Error of the Estimate	Statistics of Changes				
					Change of Square of R	Change of F	df1	df2	Next Change in F
1	0.725a	0.525	0.518	6.256	0.525	74.720	8	541	0.000

**Table 4 children-10-01264-t004:** Logistic regression.

	B	Standard Error	Wald	gl	Sig.	Exp (B)	95% CI EXP (B)	
							Lower	Upper
REB	0.695	0.252	7.596	1	0.006	2.003	1.222	3.282
Anxiety	1.176	0.265	19.631	1	0.000	3.240	1.926	5.451
Emotional dysregulation	1.066	0.259	16.996	1	0.000	2.903	1.749	4.819
Sexual orientation	1.297	0.358	13.092	1	0.000	3.657	1.812	7.383
Family violence	−0.810	0.341	5.651	1	0.017	0.445	0.228	0.867
Alcohol and drugs at school	−0.728	0.267	7.445	1	0.006	0.483	0.286	0.815
Alcohol and drugs at home	−1.189	0.355	11.242	1	0.001	0.304	0.152	0.610
Constant	−0.328	1.211	0.074	1	0.786	0.720		

B = beta; Exp (B) = standardized beta; OR = odds ratio; CI = confidence interval; *p* = statistical confidence.

## Data Availability

The information provided in this research can be obtained upon request from the author in charge. Due to ethical considerations, the data is not accessible to the public.

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
