# Peer review of "Depression Disorders in Mexican Adolescents: A Predictive Model"

_children, 2023, doi:10.3390/children10071264_

Round 1

Reviewer 1 Report

The main question addressed by the research is to find the risk factors for the development of depressive disorder in Mexican adolescents.  The topic is relevant in the field. The results do not add relevant data. The authors should better organize and analyze the risk factors considered. The topics are not presented systematically. The references are appropriate. I have no additional comments on the tables and figures.

The idea is appreciable, but a lack of organization of the scientific premises already emerges in the introduction. The results are reported in a confusing way. The discussion needs to be better organised. The correlation between alcohol and drug consumption is binary: depressed subjects consume more alcohol and drugs.

Author Response

Since there were different responses among the reviewers, it was decided to make the changes suggested by Reviewer 2, e.g., the reviewer indicates that the introduction is very good, while Reviewer 1 indicates that it is weak; the same applies to the references. 
We hope that the changes made will be accepted by reviewer 1.

Reviewer 2 Report

Dear Authors,

The manuscript, titled "Depression Disorders in Mexican Adolescents," is related to examining the risk of depression among adolescents in Mexico retrospectively. 

- The introduction is well written, and you documented the signs, etiology, epidemiology, and effecting factors of depression. 

- Lines between 59-62 need a reference. 

- Please remember the paragraph should consist of at least three sentences, and there is some too-short paragraphs whole manuscript. 

- The abbreviation REB must be clarified in the sentences; is it Eating Behavior Disorders? 

- In the method section, you should explain the measurements in detail. Who developed or did the validity and reliability of the scale, and how was reliability (Cronbach alpha) in the current study?

- I need help understanding how you collected the data retrospectively from adolescents.

- Did the informed consent obtain from the adolescents' parents or legal guardians in this study? If yes, please add. 

- In the results section, please add the normality test or Kurtosis, Skewness of the scale and explain why you chose the parametric tests. 

- Please add the reference for the correlation because 0.25 and 0.50 are weak, and 0.50-0.75 is moderate according to some references. 

- How did you establish the regression model? Only literature or you found a statistical difference in groups such as drug and alcohol consumption? Please explain. 

- In the discussion part, you summarize the results and add literature related to the findings. It needs your comment on these findings and your recommendation for preventing depression among adolescents. 

I wish you success in your work. 

Round 2

Reviewer 1 Report

The authors have improved the work

Author Response

Reviewer 1 made no comment

Reviewer 2 Report

Dear authors,

Thank you for the revision. This study design is not retrospective; it is cross-sectional. There needs to be more information about measurements. The scales should be introduced and referenced. The method section with references should include Kurtosis, skewness, and correlation, not the results section. The discussion part is the same as a preview version of the manuscript; I could not find the recommendation for future studies and the authors' comments. Please highlight the revision with track changes since the edit is hard to find. 

 I wish you success in your work. 

Author Response

Request Reviewer 2

In light of the suggestions made by reviewer 2, several revisions were made. Firstly, the study design was appropriately adjusted. Additionally, the scales utilized in the research were derived from the Online Test for Self-Screening: Risk Factors of Eating Disorders, Depression, Social Anxiety, and Self-Injury (OTESSED) [35]. The present article includes the alpha and omega values for each scale. Furthermore, the Statistical Analysis section now incorporates the inclusion of skewness, kurtosis, and correlation. In terms of the Theoretical Framework, information regarding alcohol and depression has been integrated. This new information has been highlighted in blue to facilitate easy identification. Furthermore, the discussion section also covers this additional content.

Round 3

Reviewer 2 Report

Thank you for revision. 

Author Response

Reviewer number two made no comments